# SimLabel: Consistency-Guided OOD Detection with Pretrained Vision-Language Models

## Abstract

Detecting out-of-distribution (OOD) data is crucial in real-world machine learning applications to prevent severe errors, particularly in safety-critical domains. Existing methods often leverage language information from vision-language models (VLMs) to enhance OOD detection by improving confidence estimation through rich class-wise text information. However, these methods primarily focus on obtaining OOD scores based on the similarity of the new sample to each in-distribution (ID) class, overlooking the OOD scores to a group of similar classes. We assume that an ID sample should consistently receive a high similarity score across similar ID classes. This paper investigates the ability of image-text comprehension among different semantic-related ID labels in VLMs and proposes a novel post-hoc strategy called SimLabel. SimLabel enhances the separability between ID and OOD samples by establishing a more robust image-class similarity metric that considers consistency over a set of similar class labels. Extensive experiments demonstrate the superior performance of SimLabel on various zero-shot OOD detection benchmarks, underscoring its efficacy in achieving robust OOD detection.

## 1 Introduction

Handling out-of-distribution (OOD) data is a critical challenge in real-world machine learning applications, particularly in safety-related domains such as autonomous driving systems and medical diagnosis Hendrycks & Gimpel (2016). Traditional image domain OOD detection methods primarily focus on visual inputs Hendrycks et al. (2020); Hsu et al. (2020); Jin et al. (2022); Shen et al. (2021); Xu et al. (2021) and develop various scoring functions Wang et al. (2022); Hendrycks & Gimpel (2016) to distinguish OOD data from in-distribution (ID) classes. Due to the unimodal nature of these approaches, they rely solely on visual information, limiting their ability to leverage rich semantic information inherent in text labels.

The emergence of Vision-Language Models (VLMs), notably CLIP Radford et al. (2021), has opened new opportunities to leverage paired image and text information for OOD detection. For instance, ZOC Esmaeilpour et al. (2022) utilizes a trainable captioner to generate OOD labels and introduces the task of Zero-Shot OOD detection, which does not require training on ID samples. Maximum Concept Matching (MCM) Ming et al. (2022) proposes a distance-based zero-shot OOD detection method where the fundamental assumption is that images are more likely to be ID if their embeddings are closer to ID text embeddings, and vice versa. However, the naive textual prompt construction in this method neglects the rich semantic textual information of the ID classes, leading to less effective ID/OOD separation.

To address these limitations, variants of the MCM score have been presented. For instance, Dai et al. (2023) introduces class-wise attributes to enhance the confidence score between ID images and labels, providing more accurate and expressive descriptions for improved performance. Similarly, Wang et al. (2023) introduces and trains a negative prompt for each ID class using an external dataset, performing OOD detection by combining scores from both negative and traditional prompts. However, these methods primarily focus on learning individual class-wise textual information, overlooking the semantic information existing among different classes. In Fig. 1 (a), we show OOD detection results for ID (top) and OOD (bottom) samples, respectively, without considering intra-class similarity. The inaccurate predictions over OOD samples motivate our investigation into aggregating OOD scores

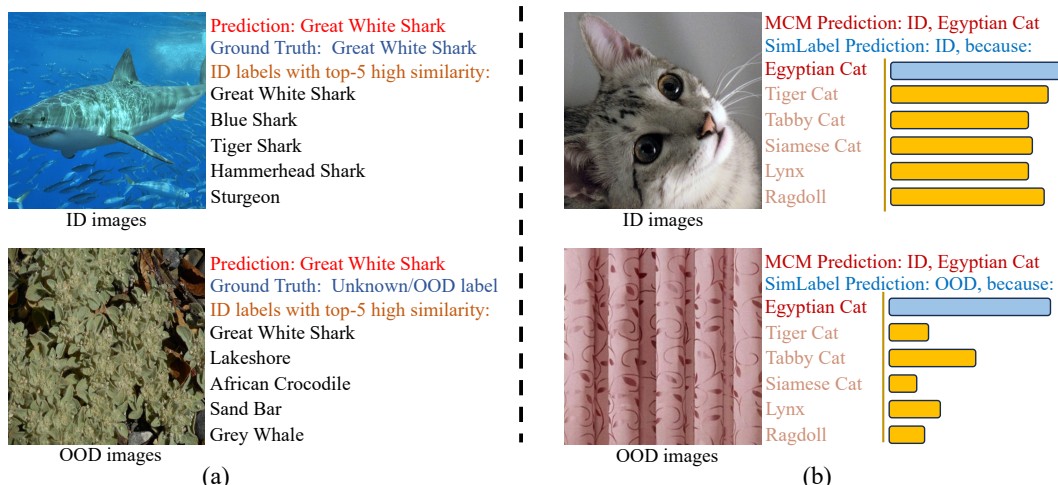

Figure 1: (a) Illustration of VLMs guided OOD detection for ID (top image from ImageNet Deng et al. (2009)) and OOD (bottom image from iNaturalist Horn et al. (2018)) samples, respectively. (b) Comparison between the proposed SimLabel and the baseline MCM Ming et al. (2022) for ID (top) and OOD (bottom) samples, demonstrating how our method detects by aggregating OOD scores across similar ID labels (yellow and blue bars denote image & similar-classes-labels similarity and image & class-labels similarity respectively).

for similar classes to determine OOD. Our basic assumption is that an ID sample should consistently have high similarity scores across similar ID classes. This assumption motivates us to devise a new method for detecting OOD samples based on measuring the consistency among semantically related labels from the ID classes.

To further illustrate our motivation, for every ID label, we introduce a set of accompanying or similar labels from either the ID labels or external knowledge that enable the model to detect OOD samples in a second direction: consistency between image and similar classes. Specifically, as demonstrated in Fig. 1 (b), given ID (top) and OOD (bottom) images which are predicted as the same class by the baseline method, ID images show consistent higher similarity to the set of semantically similar ID classes than OOD images. Based on this observation, we propose SimLabel, a post-hoc method with a well-designed OOD score to detect OOD images by examining the consistency of high-similarity over similar classes. For instance, Fig. 1 (b) illustrates the detection on the OOD sample as it receives diverse similarity scores for similar classes, namely Tiger Cat, Tabby Cat, Siamese Cat etc. The proposed OOD score combines knowledge from the prediction and its similar classes (see example in Appendix D), thus better leveraging the VLMs' capabilities of comprehending class prototypes.

Additionally, we design several algorithms for selecting high-quality similar labels from the ID class or external knowledge. The choice of similar classes can be generated from three directions: text-hierarchy, world knowledge, and pseudo-image-label alignment. Extensive experiments validate that our proposed method, SimLabel, achieves superior performance across various zero-shot OOD detection benchmarks.

We summarize our main contributions as follows:

- We propose a novel post-hoc framework, SimLabel (see Sec. 3.2), that constructs the affinity between images and class prototypes with semantic-related labels for robust OOD detection.

- We introduce different and comprehensive strategies (see Sec. 3.1) for selecting similar labels from the various perspectives and illustrate the influence on OOD detection performance with the different choices of similar classes (see Sec. 4).

- We present in-depth empirical analysis, offering insights into the effectiveness of the SimLabel score (see Sec. 5) and show that SimLabel learns a robust and discriminative image-class matching score, potentially improving visual classification ability.

## 2 PRELIMINARIES

Let $\mathcal{X} = \mathcal{X}_{ID} \cup \mathcal{X}_{OOD}$, $\mathcal{L} = \{l_1, \ldots l_I\}$ and $\mathcal{P} = \{\mathrm{prompt}(l_i) \mid l_i \in \mathcal{L}\}$ be the set of images, ID labels and corresponding prompts respectively where $I$ indicates the number of ID classes and the function $\mathrm{prompt}(l_i)$ denotes the prompt template, e.g., "A photo of <label>". We define ID images as $x_{ID} \in \mathcal{X}_{ID}$ and OOD images as $x_{OOD} \in \mathcal{X}_{OOD}$.

**CLIP model Radford et al. (2021).** Given any images $x \in \mathcal{X}$ and label $l_i \in \mathcal{L}$, along with frozen text and image encoder $f_T : \mathcal{X} \to \mathcal{R}^D$ and $f_I : \mathcal{P} \to \mathcal{R}^D$ from the CLIP model, the visual features $\mathbf{h} \in \mathcal{R}^D$ and the textual feature $\mathbf{e}_i \in \mathcal{R}^D$ can be extracted as:

$$\mathbf{h} = f_I(x), \mathbf{e}_i = f_T(\mathrm{prompt}(l_i)) \tag{1}$$

where $D$ denotes the dimension of features. The CLIP model performs prediction through the measurement of cosine similarity between image embedding features $\mathbf{h}$ and text embedding features $\mathbf{e}$. Thus, the prediction can be selected as the label $\hat{l}$ with highest similarity $\mathcal{M}(x, \hat{l})$, expressed as:

$$\hat{l} = \arg\max_{l_i \in \mathcal{L}} \{\mathcal{M}(x, l_i)\} \qquad \mathcal{M}(x, l_i) = \cos(f_I(x), f_T(\mathrm{prompt}(l_i))). \tag{2}$$

**Score function.** Score function plays an essential role in OOD detection tasks. Given score function $S(\cdot)$ along with the threshold $\tau$, following many representative works in OOD detection Hendrycks & Gimpel (2016); Ge et al. (2023); Ming et al. (2022), an image $x$ can be decided as ID or OOD based on function $G$:

$$G_\tau(x) = \begin{cases} \mathrm{ID} & S(x) \geq \lambda \\ \mathrm{OOD} & S(x) < \lambda \end{cases}, \tag{3}$$

The performance of OOD detection is highly related to the design of function $S(\cdot)$, where ID samples are expected to receive higher scores than the OOD samples.

**Problem set-up.** VLMs bridge image and text modalities through a pair-matching training strategy. Thus, given a pre-trained CLIP-like Radford et al. (2021) model and pre-defined label names, one can conduct visual classification without training (namely zero-shot classification task). In this paper, we follow the setting of this task, aim to develop a score in detecting any input label which not belong to any class without sacrificing the classification accuracy.

## 3 METHODOLOGY

In estimating confidence for determining whether images are ID or OOD, prior works such as the MCM score and Ming et al. (2022); Dai et al. (2023) are confined to their predicted label $l_i$. However, VLMs are trained on extensive datasets Radford et al. (2021) and have demonstrated their image-text comprehension ability by measuring the similarity between images and various text embedding. Thus, as illustrated in Fig. 1, this confinement to a single label prevents the model from fully capturing the image-text comprehension capabilities of VLMs, ultimately compromising the model's confidence estimation on ID samples.

The motivation behind our method is to enhance the model's robustness by extending the label set for each class, effectively utilizing the VLMs' image-text comprehension capabilities. Thus, we demonstrate several intuitive algorithms for generating similar classes to extend each ID class (Sec. 3.1). We then develop a post-hoc SimLabel score for OOD detection by measuring the consistency between images & similar-class-labels similarity (Sec. 3.2).The pipeline of estimating SimLabel score in detecting OOD samples can be seen in Fig. 2.

### 3.1 SIMILAR CLASS GENERATION

**Overview of similar classes generation.** We aim to generate a set of labels for each class $l_i$ that have higher affinity/similarity to ID samples compared with OOD samples. We then refer these labels as similar labels $\mathcal{D}(l_i)$ for each class $l_i$. In this section, we propose three methods for generating similar classes: **1.** exploring the text hierarchy among class labels and select the labels under the same super-class as similar classes (see Sec. 3.1.1); **2.** using the external large language models/world knowledge in generating similar classes (see Sec. 3.1.2); **3.** utilizing the similarity between ID images and ID labels for selecting the similar classes (see Sec. 3.1.3).

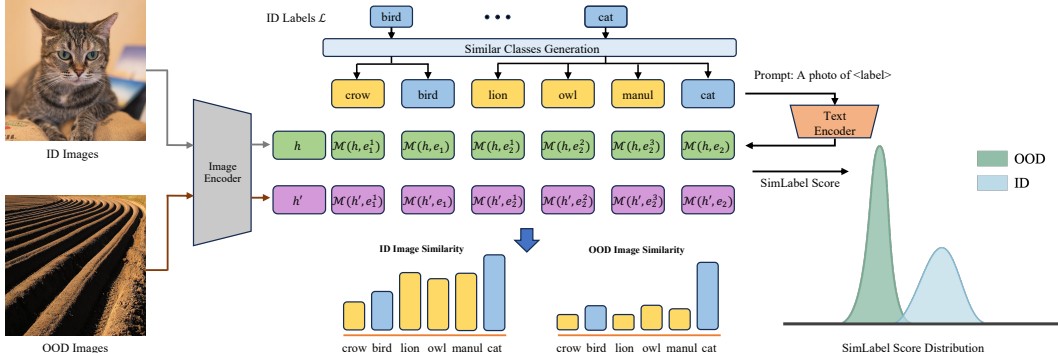

Figure 2: **Overview of the SimLabel zero-shot OOD detection framework.** The image encoder first encodes ID and OOD images into image embeddings $\mathbf{h}$ and $\mathbf{h}'$, respectively. For every class label (represented as blue blocks) in the ID label set $\mathcal{L}$, similar classes (represented as yellow blocks) are generated through the process of similar class generation defined in Sec. 3.1. The text encoder extracts ID and their similar class labels into text embeddings with prompts and the image-text similarities are measured using the function defined in Eq. 2. Both image and text encoders are frozen. The below charts indicate that, ID *cat* images, compared with OOD images that are predicted into the *cat*, will produce higher similarity to similar classes such as *lion, owl, manul*. Our proposed SimLabel (detailed in Sec. 3.2) conducts OOD detection by utilizing image & class-label and image & similar-classes-label similarity.

### 3.1.1 SIMILAR CLASSES BASED ON TEXT HIERARCHY

Utilizing inherent hierarchical information among different class labels is a natural idea in clustering ID classes into groups with high semantic correlation and Novack et al. (2023) has shown a precise construction of tree-structured hierarchical label sets over various datasets. Here, we follow the pipeline shown in Novack et al. (2023) to build hierarchical label sets among ID labels where the subsets of the hierarchy can be seen in Appendix B. For each class $l_i$, we select the class labels under the same super-class as the set of similar classes $\mathcal{D}(l_i)$. We denote the SimLabel score with these hierarchical similar classes as SimLabel-H.

### 3.1.2 SIMILAR CLASSES FROM LARGE LANGUAGE MODELS

Nevertheless, the above method may face with the problem of unbalanced ID label space which result in the lack of similar labels for some rare ID classes. Large language models (LLMs), such as GPT-3 Brown et al. (2020), possess extensive world knowledge across various domains, making us a direct approach for generating similar classes. We prompt LLMs to generate similar classes $\mathcal{D}(l_i)$ where they share similar visual features. We randomly select several visual categories and manually compose similar classes using a one-shot in-context example, and the details of the templates for prompting LLMs have been shown in Appendix A. An example of generated similar classes is listed in Fig. 3.

Additionally, although the LLM generates semantically correlated similar classes from world knowledge, we find that some classes are more semantic-related and potentially affect the OOD detection performance. We measure the semantic textual similarity between $l_i$ and every similar label $\mathbf{d} \in \mathcal{D}(l_i)$ with cosine similarity in feature space, and selecting the labels with top-k similarities as the final set of similar classes. We denote the SimLabel score with prompting LLMs as SimLabel-L.

### 3.1.3 SIMILAR CLASSES WITH IMAGE-TEXT ALIGNMENT

Selection of similar classes is to find labels whose ID samples have high affinity. The above two selection methods are limited to the textual modality, where the sensitivity of the text encoder Miyai et al. (2023) may result in the incorrect selection of similar classes. In this section, we introduce a new strategy for selecting similar classes with consideration of ID image-text alignment. For each ID image, we can first perform zero-shot visual classification to assign it to a pseudo ID class and select the labels with top similarities as similar classes.

Figure 3: This figure illustrates samples of similar classes for the class "Great White Shark" using methods in Sec. 3.1.3 and Sec. 3.1.2. **Left** The similar classes generated from the ID labels. **Right** The similar classes generated by LLM.

The generation of similar classes through single-image-text alignment can be problematic due to CLIP's inaccuracy and contingency in image-text alignment. To address this issue, we propose a robust method to select similar classes that consistently show high similarity among most ID samples $\mathcal{X}_i$ predicted into class $l_i$. Specifically, our assumption is that, for every image $x_i^j \in \mathcal{X}_i$, set of similar class (donates $\mathcal{D}(x_i^j)$) with top-k similarity varies but the true similar classes representing class prototype $l_i$ will consistently or highly possibly show in $\mathcal{D}(x_i^j)$. In this case, we record all set of similar labels $\mathcal{D}(x_i^j)$ for each images in $\mathcal{X}_i$ and select the labels with top occurrence among all sets as the similar classes $\mathcal{D}(l_i)$ for class prototype $l_i$. The detail of algorithm is shown in Appendix C. We denote the SimLabel score by referring to image-text alignment as SimLabel-I.

## 3.2 OOD DETECTION WITH SIMILAR CLASSES

**SimLabel Score.** With the generation of high-quality similar-classes, in this section, we propose our pipeline in using image & similar-class-label similarity for detecting OOD sample as illustrated in Fig. 2. For every class, given the class-wise similar classes $\mathcal{D}(l_i)$ generated in Sec. 3.1, we merge them with the class label $l_i$ to obtain an extended class-wise label set. Then, the extended labels set are fed into the text encoder to obtain text embeddings as shown by the yellow bar in Fig. 2 and CLIP calculates the cosine similarities between the text and image embeddings. Formally, we define the affinity $\mathcal{A}(x, l_i)$ between images $x$ and class $l_i$ as:

$$\mathcal{A}(x, l_i) = \mathcal{M}(x, l_i) + \alpha * \sum_{d \in \mathcal{D}(l_i)} \mathcal{M}(x, d)/|\mathcal{D}(l_i)| \tag{4}$$

where $|\mathcal{D}(l_i)|$ indicates the cardinality of similar classes and $\alpha$ is a hyper-parameter that determines the weight of image & similar-classes-label similarity. The higher the $\alpha$ is, the impact of similar classes in SimLabel score will be amplified. Intuitively, Eq. 4 enhances the estimation of connections between image and class prototype with the combination of image & class-label and weighted image & similar-classes-label similarity. Motivated by the assumption in Ming et al. (2022) that the maximum similarity of ID image-text alignment shows advantages over OOD samples, we formally define our SimLabel score with the maximum matching score as:

$$S(x; \mathcal{L}, \tau) = \max_{l_i \in \mathcal{L}} \frac{e^{\mathcal{A}(x, l_i)/\tau}}{\sum_i^I e^{\mathcal{A}(x, l_i)/\tau}} \tag{5}$$

where $\tau$ indicates the temperature scalar. Our OOD detection function can then be formulated as:

$$G(x; \mathcal{L}, \tau) = \begin{cases} \text{ID} & S(x; \mathcal{L}, \tau) \geq \lambda \\ \text{OOD} & S(x; \mathcal{L}, \tau) < \lambda \end{cases}, \tag{6}$$

where $\lambda$ is chosen so that a high fraction of ID data (e.g., 95%) is above the threshold. For sample $x$ that is classified as ID, one can obtain its class prediction based on the nearest prototype: $\hat{y} = \arg\max_{l_i \in \mathcal{L}} \mathcal{A}(x, l_i)$.

Table 1: OOD detection performance comparison with baselines on ImageNet-1k benchmark using CLIP-B/16 model. SimLabel-H, SimLabel-L and SimLabel-I indicate the SimLabel score using similar labels generated in Sec. 3.1.1, Sec. 3.1.2, Sec. 3.1.3 respectively.

| Method | iNaturalist | | SUN | | Places | | Textures | | Average | |
|---|---|---|---|---|---|---|---|---|---|---|
| | AUROC↑ | FPR↓ | AUROC↑ | FPR↓ | AUROC↑ | FPR↓ | AUROC↑ | FPR↓ | AUROC↑ | FPR↓ |
| MSP | 77.74 | 74.57 | 73.97 | 76.95 | 74.84 | 73.66 | 72.18 | 79.72 | 74.68 | 76.22 |
| MaxLogit | 88.03 | 60.88 | 91.16 | 44.83 | 88.63 | 48.72 | 87.45 | 55.54 | 88.82 | 52.49 |
| Energy | 87.18 | 64.98 | 91.17 | 46.42 | 88.22 | 50.39 | 87.33 | 57.40 | 88.48 | 54.80 |
| ReAct | 86.87 | 65.57 | 91.04 | 46.17 | 88.13 | 49.88 | 87.42 | 56.85 | 88.37 | 54.62 |
| ODIN | 57.73 | 98.93 | 78.42 | 88.72 | 71.49 | 85.47 | 76.88 | 87.80 | 71.13 | 90.23 |
| KNN | 94.52 | 29.17 | **92.67** | 35.62 | 91.02 | 39.61 | 85.67 | 64.35 | 90.97 | 42.19 |
| MCM | 94.40 | 32.18 | 92.27 | 39.29 | 89.82 | 44.92 | 85.99 | 58.03 | 90.62 | 43.61 |
| Dai et al. | 95.54 | 22.88 | 92.60 | **34.29** | 89.87 | 41.63 | **87.71** | **52.02** | 91.43 | 37.71 |
| SimLabel-H | 94.24 | 30.06 | 89.99 | 51.07 | 86.15 | 58.62 | 81.03 | 72.09 | 87.86 | 52.96 |
| SimLabel-L | 96.15 | 19.13 | 88.40 | 50.13 | 91.42 | 45.01 | 86.57 | 56.70 | 90.64 | 42.74 |
| SimLabel-I | **96.74** | **15.28** | 90.35 | 42.84 | **93.45** | **34.07** | 87.07 | 53.65 | **91.90** | **36.46** |

## 4 EXPERIMENT

### 4.1 EXPERIMENT SETUP

**Datasets and benchmarks.** We evaluate our method on the ImageNet-1k OOD benchmark Huang et al. (2021) and primarily compare it with the MCM method Ming et al. (2022) due to its promising and consistent performance in the zero-shot OOD detection task. The ImageNet-1k OOD benchmark is a widely used performance validation method that uses the large-scale visual dataset ImageNet-1k as ID data and iNaturalist Horn et al. (2018), SUN Xiao et al. (2010), Places Zhou et al. (2016), and Texture Cimpoi et al. (2013) as OOD data, covering a diverse range of scenes and semantics. Each OOD dataset has no classes that overlap with the ID dataset.

**Implement details.** In our experiments, we adopt CLIP Radford et al. (2021) as the target pre-trained model, which is one of the most popular and publicly available VLMs. Note that our method is not limited in CLIP; it can be applicable to other vision-language pre-trained models that enable multi-modal feature alignment. Our experiments are primarily conducted using the CLIP-B/16 model, which consists of a ViT-B/16 Transformer as the image encoder and a masked self-attention Transformer Vaswani et al. (2017) as the text encoder. For selecting similar classes in SimLabel-H, we follow the construction of hierarchical label sets in Novack et al. (2023) to obtain accurate super-classes for the ImageNet labels and generate similar classes. The LLMs we prompt for SimLabel-L is GPT-4 Achiam et al. (2023). In generating similar classes for SimLabel-L and SimLabel-I score, we select the quantity of similar classes $k = 6$. Additionally, following the theoretical analysis and setting in Ming et al. (2022), we set temperature $\tau = 1$. We set the weight of image & similar-classes-label $\alpha = 1$. All experiments are conducted on a single NVIDIA 4090 GPU.

**Metric.** For evaluation, we mainly use two metrics: (1) the false positive rate (FPR@95) of OOD samples when the true positive rate of in-distribution samples is 95%.(2) the area under the receiver operating characteristic curve (AUROC).

### 4.2 EXPERIMENT RESULT AND ANALYSIS

**Comparison with baselines.** We conduct comprehensive OOD evaluation on the ImageNet-1k benchmark. We compare our proposed method SimLabel with other existing OOD detection methods in Table 1. The methods we compare can be divided into two categories: uni-modal OOD methods and multi-modal OOD methods. Specifically, the methods we compare include various multi-modal OOD methods based on MCM Ming et al. (2022); Dai et al. (2023) and several traditional uni-modal OOD detection methods including MSP Hendrycks & Gimpel (2016), MaxLogit Hendrycks et al. (2022), Energy Liu et al. (2021), ReAct Sun et al. (2021), ODIN Liang et al. (2020) and KNN Sun et al. (2022). Notably, we demonstrate three methods in establishing the SimLabel Score for OOD detection, namely SimLabel-H, SimLabel-L, SimLabel-I, which represent the three distinct

Table 2: OOD detection results on various fine-grained datasets comparing with MCM where ID dataset is CUB-200Welinder et al. (2010), Food-101Bossard et al. (2014), Oxford-IIIT PetParkhi et al. (2012) and Stanford CarsKrause et al. (2013).

| ID Dataset | Method | iNaturalist | | SUN | | Places | | Textures | | Average | |
|---|---|---|---|---|---|---|---|---|---|---|---|
| | | AUROC↑ | FPR↓ | AUROC↑ | FPR↓ | AUROC↑ | FPR↓ | AUROC↑ | FPR↓ | AUROC↑ | FPR↓ |
| CUB200 | MCM | 98.43 | 8.68 | 99.07 | 4.94 | 98.59 | 6.45 | 99.05 | 4.70 | 98.79 | 6.19 |
| | SimLabel-I | **99.50** | **2.25** | **99.49** | **2.72** | **99.20** | **3.50** | **99.79** | **0.80** | **99.49** | **2.32** |
| Food101 | MCM | 99.39 | 1.81 | 99.31 | 2.71 | 99.07 | 4.01 | 98.03 | 6.13 | 98.95 | 3.67 |
| | SimLabel-I | **99.54** | **0.98** | **99.42** | **2.11** | **99.28** | **2.90** | **98.22** | 5.30 | **99.12** | **2.82** |
| Pets | MCM | 99.32 | 2.78 | 99.75 | 0.93 | 99.65 | 1.62 | 99.78 | 1.01 | 99.62 | 1.59 |
| | SimLabel-I | **99.65** | **0.57** | **99.93** | **0.03** | **99.81** | **0.46** | 99.66 | **0.76** | **99.76** | **0.46** |
| Cars | MCM | 99.79 | 0.09 | **99.97** | **0.02** | 99.89 | 0.30 | **99.97** | **0.02** | 99.90 | 0.11 |
| | SimLabel-I | **99.86** | **0.02** | 99.94 | 0.04 | 99.87 | 0.33 | 99.96 | **0.02** | **99.91** | **0.10** |

Table 3: **Zero-shot OOD detection performance comparison on hard OOD detection tasks.** Following the MCM Ming et al. (2022), we use the subsets of ImageNet-1kDeng et al. (2009) (ImageNet-10 and ImageNet-20) for testing the performance of SimLabel on hard OOD detection task.

Table 4: **Zero-shot OOD detection with various VLM architectures other than CLIP.** We use the average performance of ImgeNet-100 (ID) vs. four common OOD datasets: iNaturalist Horn et al. (2018), SUN Xiao et al. (2010), Places Zhou et al. (2016), and Texture Cimpoi et al. (2013).

| ID dataset | OOD dataset | Method | AUROC↑ | FPR95↓ |
|---|---|---|---|---|
| ImageNet-10 | ImageNet-20 | MCM | 98.71 | 5.00 |
| | | SimLabel-I | **99.30** | **3.20** |
| ImageNet-20 | ImageNet-10 | MCM | 97.88 | 17.40 |
| | | SimLabel-I | **98.43** | **12.00** |

| Architecture | Method | AUROC↑ | FPR95↓ |
|---|---|---|---|
| AltCLIP | MCM | 83.40 | 71.70 |
| | SimLabel-I | **84.66** | **65.13** |
| GroupViT | MCM | 69.45 | 82.38 |
| | SimLabel-I | **73.57** | **80.38** |

similar classes generation methods shown in Sec. 3.1 respectively. The introduction of SimLabel-I score produces the best performance followed by the SimLabel-L while it is worth mentioning that SimLabel-H and SimLabel-I both generate similar classes with refer to ID classes while their OOD performance varies. In Sec. 5.3, we analyze the limitation of our method which explains the failures in SimLabel-H. On average, as a post-hoc method, our SimLabel-I, using the CLIP model with ViT-B-16 and similar classes generated by image-text alignment, demonstrates significant enhancements of 0.47% and 1.25% in terms of AUROC and FPR95 concerning formal Dai et al. (2023).

**OOD detection of SimLabel on fine-grained datasets.** Following the setup from MCM, we also explore the performance of SimLabel on small fine-grained datasets demonstrating our method's ability in superior generalizability, consistently achieving remarkable average performance for zero-shot OOD detection tasks of any scale. Specifically, we uses CUB200 Welinder et al. (2010), Food-101 Bossard et al. (2014), Oxford-IIIT Pet Parkhi et al. (2012), and Stanford Cars Krause et al. (2013) and ID dataset and iNaturalist Horn et al. (2018), SUN Xiao et al. (2010), Places Zhou et al. (2016), and Texture Cimpoi et al. (2013) as OOD data. The result is listed in Table 2, where all the experimental results are evaluated using CLIP model based on ViT-B-16. Our proposed method SimLabel-I demonstrates significant improvement in OOD detection on most fine-grained ID datasets and OOD datasets.

**OOD detection of SimLabel on hard OOD detection tasks**. Following MCM's Ming et al. (2022) setting, we aims to investigate the performance of our method SimLabel-I on hard OOD detection tasks where OOD samples that are semantically similar to ID samples Winkens et al. (2020) are particularly challenging for OOD detection algorithms as shown in Table 3. Notably, because MCM dose not provides details on generating spurious OOD samples, we mainly test SimLabel-I on semantic-hard OOD detection task. Thus, we alternate using ImageNet-10 and ImageNet-20 as ID and OOD data for testing SimLabel-I. The result demonstrated in Table 3 indicates that our method SimLabel-I significantly outperform MCM in hard-OOD detection task indicating SimLabel's superior ability on distinguishing semantic-hard OOD samples.

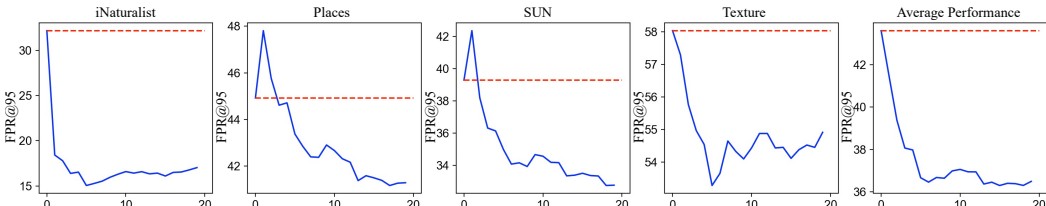

Figure 4: This figure illustrates how the FPR@95 changes with different choices on the number of similar classes ($k$) for each class in SimLabel-I Score on ImageNet-1k benchmarkHendrycks & Gimpel (2016). The x-axis and y-axis are the $k$ values and FPR@95 performance respectively. Additionally, the red dashed line is the MCM score result serving as a baseline for comparison.

### 4.3 Ablation study

**A small set of similar classes is sufficient.** We provide an empirical study to show that when generating similar classes, a small set of similar classes is sufficient. The similar classes generation mentioned in Sec. 3.1 are based on selecting labels with top-k similarities. If k is too small, some representative classes may be overlooked, while a high k value may result in the selection of non-semantically related labels (see Fig. 3). Here, we empirically analyze the effectiveness of the number of similar classes in Fig. 4. On average, a small set of similar classes within each class can significantly improve the effectiveness of OOD detection performance compared to the conventional MCM score. However, as the number of similar classes increases, the improvement in OOD detection tends to plateau. This suggests that the small size of similar classes where the k is chosen as 6 (where the inflection occurs) is sufficient to enhance OOD detection performance.

**The weight of image & similar-class-label similarity should be moderate.** The class label $l_i$ plays a dominant role in representing class prototype and we provide an empirical study to show that the weight of image & similar-class-label similarity should be moderate. We vary the hyper-parameter $\alpha$ within a wide range in Eq. 4 and conduct the OOD detection with SimLabel-I score. The experiment uses ImageNet-1k as ID and iNaturalist Horn et al. (2018), SUN Xiao et al. (2010), Places Zhou et al. (2016), and Texture Cimpoi et al. (2013) as various OOD datasets. The average performance over four OOD datasets is shown in Table 5. The results indicate that either excessive emphasis on similar classes or their disregard can hinder OOD detection. Therefore, a moderate choice of $\alpha$ can significantly enhance OOD detection. The optimal weight is found to be around $\alpha = 1$, indicating roughly equal contributions from image & class-label similarity and image & similar-classes-label affinity.

**SimLabel on various VLM architectures** We conduct our main experiments based on CLIP-B/16 Radford et al. (2021) while it is important to verify how SimLabel works on various VLM architectures. We provides additional experiments in investigating the effectiveness of SimLabel based on various VLM architectures including AltCLIP Chen et al. (2022) and GroupViT Xu et al. (2022) which are two common-used VLM architectures. The experiment result is shown in Table 4 and we mainly compare with MCM Ming et al. (2022) score. The result indicates the our method SimLabel-I significantly outperforms MCM based on various VLM architecture which shows that our method is model-agnostic.

## 5 A closer look at SimLabel

### 5.1 SimLabel builds a robust and discriminative image-class pairing

Maintaining high ID classification accuracy is one important factor in measuring the effectiveness of our method according to the problem set-up mentioned in Sec. 2. SimLabel enhances the separability between ID and OOD data by building a robust image-class pairing mechanism with the help of consistency measurement over similar classes. A natural question is whether this multi-label affinity would hinder the model's ability to learn discriminative image-class pairing. We aim to answer this question by conducting zero-shot visual classification with SimLabel.

Table 5: **The influence of $\alpha$ on SimLabel-L and SimLabel-I.** We use the average performance of ImgeNet-1k (ID) vs. four common OOD datasets: iNaturalist Horn et al. (2018), SUN Xiao et al. (2010), Places Zhou et al. (2016), and Texture Cimpoi et al. (2013).

Table 6: **Zero-shot visual classification performance on various datasets.** We demonstrate the accuracy of SimLabel-I and CLIP-B/16 in doing prediction over several common datasets cub200 Welinder et al. (2010), ImageNet Deng et al. (2009) and ImageNetV2 Recht et al. (2019)

| $\alpha$ | 0 | 0.1 | 0.5 | 1 | 5 | 10 | 100 |
|---|---|---|---|---|---|---|---|
| SimLabel-I | 43.61 | 41.63 | 37.73 | 36.46 | 40.27 | 42.05 | 50.92 |
| SimLabel-L | 43.61 | 42.24 | 41.13 | 42.74 | 51.70 | 54.58 | 62.83 |

| Datasets | ImageNet | ImageNetV2 | cub200 |
|---|---|---|---|
| CLIP-B/16 | 66.60 | 60.61 | 55.71 |
| SimLabel-I | 67.88 | 61.29 | 57.56 |

As analyzed in Sec. 3.2, given any ID image $x \in \mathcal{X}$ and label $\mathcal{L}$, one can obtain the class prediction based on Eq. 4 with SimLabel Score. Thus, we conduct the zero-shot visual classification based on SimLabel score on various datasets including a large-scale dataset ImageNet, ImageNetV2 Deng et al. (2009); Recht et al. (2019) and small-scale fine-grained dataset: CUB-200 Welinder et al. (2010). The results, presented in Table 6, indicate that our method, which pairs images with similar class labels, improves ID classification accuracy compared to traditional zero-shot visual classification shown in Eq. 2. This demonstrates that our method, SimLabel, enables a more robust and discriminative image-class prototype matching, thereby enhancing ID/OOD separability and maintaining high ID classification accuracy.

## 5.2 THE EFFECTIVENESS OF ISOLATED IMAGE & SIMILAR-CLASSES-LABEL AFFINITY.

Our method, namely the SimLabel score, assumes that when measuring similarity between ID samples and ID labels set, other ID labels also show high similarity beyond the ground truth class label $l_i$. Ideally, the affinity with only the image & similar-classes-label should still be representative of an image-class pairing even without the class label $l_i$. We verify this assumption by reformulating the image-class prototype similarity in Eq. 5 to include only image and similar-classes-label matching: $\mathcal{A}(x, l_i) = \sum_{d \in \mathcal{D}(l_i)} \mathcal{M}(x, d)/|\mathcal{D}(l_i)|$. If the assumption does not hold, the similarity between the ID image and other ID labels would be identical, rendering the design of image-class affinity meaningless.

Here, we use this affinity to conduct OOD detection with similar classes generated for SimLabel-I on ImageNet benchmark. The results are shown in Table 7 (denoted as SimLabel-S). Although its performance is not as good as SimLabel-I, its OOD performance still demonstrates the ability to represent a class prototype without a class label. The result verifies the assumption in Fig. 1 and provides strong support for the design of SimLabel.

## 5.3 LIMITATION AND FUTURE WORK

The proposed method relies on a "similar class pool" to decide robust OOD score. One of the basic requirements of our method is that the distribution of ID class should be relatively uniform, or balanced. For the long-tailed scenarios, it is harder to select informative similar classes for the tail classes than for the head ones, posing challenges for the proposed consistency-guided OOD detection method. Thus, the non-uniform quantity of class-wise similar classes hinders the measurement of image-class affinity in terms of consistency, explaining the failure cases when using SimLabel-H on zero-shot OOD detection. One potential solution is extending the tail class by introducing extra sibling or child classes. Notably, in expanding the classes pool, a well-designed classes selection is needed for its potential problem of bring noiseDai et al. (2023) explaining the failure of SimLabel-L. Another limitation of our work stems from the design of image & similar-classes-label similarity in Eq. 4 where we assume the equal contribution for each similar class $d \in \mathcal{D}(l_i)$. In real-world scenarios, the semantic similarity between two classes varies so the uniform weights on every similar class result in the limited utilization of semantic information across the similar classes. An extension of Eq. 4 can be investigated by accurately measuring class-wise similarity within the similar class pool.

Table 7: Zero-shot OOD detection of SimLabel-S and SimLabel-I on ImageNet-1k benchmark following MCM Ming et al. (2022).

| Method | iNaturalist | | SUN | | Places | | Textures | | Average | |
|---|---|---|---|---|---|---|---|---|---|---|
| | AUROC↑ | FPR↓ | AUROC↑ | FPR↓ | AUROC↑ | FPR↓ | AUROC↑ | FPR↓ | AUROC↑ | FPR↓ |
| SimLabel-I | 96.74 | 15.28 | 90.35 | 42.84 | 93.45 | 34.07 | 87.07 | 53.65 | 91.90 | 36.46 |
| SimLabel-S | 95.13 | 25.50 | 86.88 | 56.29 | 90.59 | 50.84 | 82.51 | 60.30 | 88.78 | 48.23 |

# 6 RELATED WORKS

**Out-of-Distribution Detection.** Conventionally, the objective of OOD detection is to derive a binary ID-OOD classifier to detect OOD images within the test dataset and designing the OOD score is one of the most important tasks in OOD detection. The design of OOD score can be divided into three main categories: probability-based, logit-based, and feature-based. MSP Hendrycks & Gimpel (2016) uses the maximum predicted probability as the score and Liang et al. (2020) aims to get rid of the over-confidence through perturbing the inputs and re-scaling the logits. MaxLogits Hendrycks et al. (2022) utilizes the maximum of logits as the score and Energy Liu et al. (2021) defines the energy-function as the OOD score. ReAct Sun et al. (2021) and DICE Sun & Li (2022) further investigate the improvement of energy score through the feature clipping and discarding. For the feature-based method, Lee Lee et al. (2018) propose the score via the measurement of minimum Mahalanobis distance between the feature and the class-wise centroids as the OOD score. KNN Sun et al. (2022) investigates the effectiveness of non-parametric nearest-neighbor distance for detecting OOD samples.

**OOD Detection with Vision-Language Representations.** With the rise of large-scale pre-trained VLMs, there are various works focusing on utilizing textual information for visual OOD detection. Fort et al.Fort et al. (2021) firstly propose the utilization of VLM for OOD detection through the generation of the candidate OOD labels. MCM Ming et al. (2022) is a conventional post-hoc zero-shot method that uses the maximum predicted softmax value as the OOD score for OOD detection. Based on MCM, NPOS Tao et al. (2023) conducts the OOD data synthesis and fine-tunes the image encoder to find a decision boundary. Dai et al. Dai et al. (2023) introduce class-wise attributes to enhance the confidence score between ID images and labels while overlooking the semantic information among different ID labels. CLIPEN Dai et al. (2023) introduces the positive and negation-semantic prompts in separating ID and OOD domain and NegLabel Jiang et al. (2024) introduces external negative labels in enhance ID/OOD separation. Both CLIPEN and NegLabel utilize the external text knowledge in boosting OOD detection tasks, while our method aims in building a more reasonable and robustness ID image-text pairing. Our method, SimLabel, detecting samples through measuring consistency between images and similar classes, enables the model to build affinity from images to various ID labels in an interpretable and robust manner.

# 7 CONCLUSION

This paper introduces a simple and effective post-hoc method for multi-modal zero-shot OOD detection called SimLabel. It generates a set of class-wise similar labels that exhibit semantic relation to each class. Different from the naive textual prompt construction strategy in the existing VLMs-based solutions that decide OOD score based only on similarity to one ID label, we decide OOD based on similarity to a group of similar classes. Our basic assumption is that an ID sample should consistently have high similarity scores across similar ID classes, leading to the proposed consistency-guided OOD detection. Particularly, the proposed method determines whether an image is ID or OOD by measuring and comparing its affinity towards ID labels and corresponding similar classes, where we provide three different strategies to select high-quality similar classes, namely selecting similar classes through text hierarchy, prompting LLMs and pseudo-image-text pairing. Extensive experiments on various OOD detection tasks demonstrate the effectiveness of our method.

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

## A  PROMPTS TO GPT

In Sec. 3.1.2, we prompt LLMs in generating similar classes for SimLabel-L. We randomly select several visual categories and manually compose similar classes to use as a 1-shot in-context example, and here we display our detailed prompt to LLMs in generating similar classes:

> Given a specific class label from the ImageNet-1k dataset, generate a list of visually similar class labels. These labels should represent objects or entities that could easily be mistaken for each other by an image classification model due to their appearance, yet remain distinct enough in categories to be differentiated upon closer inspection.
> Please format your output as a list, separated by commas.
> Here are some examples to illustrate how you should structure your answers:
> Given class label: CD player
> Your answer: tape player, cassette player, radio, cassette, modem, desktop computer, monitor, hard disc, remote control, loudspeaker
> Given class label: coffee mug
> Your answer: cup, coffeepot, measuring cup, espresso maker, water jug, milk can, consomme, goblet, teapot Given class label: Blacknose shark
> Your answer: Blacktip reef shark, Dusky shark, Grey reef shark, Scalloped hammerhead shark, Pacific sharpnose shark, Silvertip shark
> For the class label category, generate a list of similar confusing class labels:
> Given class label: category
> Your answer:

## B  SIMILAR CLASSES REFERRED TO TEXT HIERARCHY

Current researchNovack et al. (2023) has proposed the construction of hierarchical label sets for ID labels in various datasets, including ImageNet. We follow the construction of a hierarchical tree defined in Novack et al. (2023) and select the label under the same super-class as the similar classes. For example, an ID class "hammerhead shark", the "great white shark" and "tiger shark" will be selected as similar classes because they are all under the superclass "shark". Here, we show some of the examples in visualizing the construction in Table 8.

Table 8: Subset of ImageNet class hierarchy

| Super-class | ID-labels/child-classes |
|---|---|
| shark | hammerhead shark |
| | great white shark |
| | tiger shark |
| flower | daisy |
| | orchid |
| turtle | mud turtle |
| | terrapin |
| | box turtle |
| | sea turtle |
| domestic cat | tabby cat |
| | tiger cat |
| | Persian cat |
| | Siamese cat |
| | Egyptian cat |
| reservoir | water tower |

## C  PSEUDO CODE IN GENERATING SIMILAR CLASSES

To have a better and clear understanding on our proposed method in generating similar classes as demonstrated in Sec. 3.1.3, we demonstrate the detail in Algorithm 1.

---

**Algorithm 1** Similar Class Generation with Class-wise Image-Text Similarity

---

**Input:** ID label set $\mathcal{L}$, ID sample $x_{ID} \in \mathcal{X}_{ID}$
**Output:** Similar Classes $\mathcal{D}(l_i)$ for every $l_i \in \mathcal{L}$
1: **for** $l_i \in \mathcal{L}$ **do**
2:  $\mathcal{X}_i \leftarrow \{x_i^j \mid x_i^j \in \mathcal{X}_{ID}\}$
3:  // Select the $x_i^j$ with Eq. 2 from ID samples where the zero-shot prediction is $l_i$
4:  $\mathcal{D}(\mathcal{X}_i) \leftarrow \emptyset$
5:  **for** $x_i^j \in \mathcal{X}_i$ **do**
6:    Compute $\mathcal{D}(x_i^j)$
7:    // Finding the ID labels with top-k high-similarity for $x_i^j$
8:    $\mathcal{D}(\mathcal{X}_i)$ records the all similar classes $d(x_i^j) \in \mathcal{D}(x_i^j)$
9:  **end for**
10:  $\mathcal{D}(l_i) \leftarrow Select(\mathcal{D}(\mathcal{X}_i))$
11:  // Select the label in $\mathcal{D}(x_i)$ with top-k highest occurrence
12: **end for**
13: **return** $\mathcal{D}(l_i)$ for every $l_i \in \mathcal{L}$

---

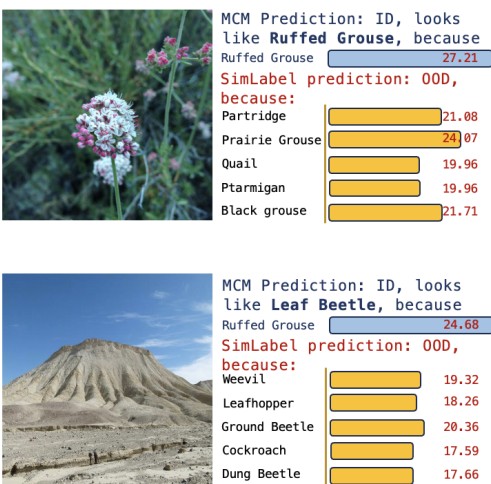

Figure 5: This figure demonstrate two practical examples on how our method SimLabel score can correct the OOD detection comparing with MCM baselineMing et al. (2022).

## D INTERPRETABLE VISUALIZATION ON SIMLABEL

In building a image-class prototype through the utilization of image & similar-classes-label provides a interpretable method in enhancing image-class connection. Here, we provides a visualization on how would this interpretability show-up in Fig. 5.

## E BROADER IMPACTS

**SimLabel Proposes a Novel, Straightforward Method in OOD Detection** In our work, we propose a straightforward idea for the task of OOD detection: an image that is correctly correlated with one class should also show high similarity to similar classes. Based on this simple motivation, we propose several direct and comprehensive approaches for selecting similar classes to use in OOD detection tasks. The effectiveness of our method, SimLabel-I, stemming from the potential of CLIP's text comprehension ability, has been verified in various experiments as shown in the experiments section and general rebuttal section.

**SimLabel Offers a New Direction for Enhancing ID/OOD Separation** In enhancing ID/OOD separation through modeling ID image-class labels, existing works Wang et al. (2023); Dai et al. (2023) either focus on a single class prototype or build an image-whole ID label setJiang et al. (2024). NegLabelJiang et al. (2024) introduces a new score utilizing synthesized negative text labels, suggesting that ID images should have higher affinity to ID text labels while showing low affinity to other texts. Building the similarity between ID images and the full-label space, low-semantic correlation may hinder understanding Jiang et al. (2024) because ID labels may vary significantly, as seen in the ImageNet-1K dataset.

Our paper proposes a novel and potential direction for building ID image-label prototypes, suggesting that ID images should show high similarity to similar labels, acting in a more interpretable way compared to Jiang et al. (2024). Therefore, our idea can be a further direction to cooperate with existing works such as CLIPEN or NegLabel, providing a more interpretable affinity between ID images and various label pools.

**SimLabel Works as a Novel Uncertainty Estimation Method** Our main method (SimLabel-I) introduces a novel strategy for uncertainty estimation over ID class prototypes and enhances ID/OOD separation through connections among ID classes.Thus, our main comparing baseline is MCM Ming et al. (2022) where the superior performance in Table 2 indicates the effectiveness of our method.

We provide a detailed analysis of our method in Section 5. The analysis of discriminative feature learning further highlights the superior performance of our work. Zero-shot visual classification using SimLabel, as shown in Table 2, indicates the superior performance of our method in building discriminative representations for each class. The improvement over CLIP-based zero-shot classification highlights the superior effectiveness of our approach.

