# OpenReview forum: "SimLabel: Consistency-Guided OOD Detection with Pretrained Vision-Language Models"
_ICLR.cc/2025/Conference — ICLR 2025 Conference Withdrawn Submission_

### Official Review · Reviewer_Dumo · 2024-10-28

**Soundness:** 2
**Presentation:** 3
**Contribution:** 2
**Rating:** 3
**Confidence:** 5

**Summary:**

This paper addresses the challenge of OOD detection, critical for preventing errors in machine learning applications, particularly in safety-sensitive areas. Existing methods using VLMs improve OOD detection by leveraging class-specific textual information. However, they focus mainly on the similarity between a new sample and each ID class, often neglecting the broader context of similar classes. To address this, the authors propose a novel approach, SimLabel, which enhances OOD detection by utilizing a robust image-class similarity metric that consistently evaluates similarity across related classes. Experimental results highlight SimLabel’s superior performance in zero-shot OOD benchmarks, underscoring its effectiveness in enhancing the separability of ID and OOD samples and promoting more reliable OOD detection.

**Strengths:**

1. The paper is well-written and easy to understand. The framework diagram and visualization results are helpful for understanding the method.
2. The paper proposes multiple feasible concrete solutions based on a single concept, providing a comprehensive methodology section.

**Weaknesses:**

1. The performance reported in this paper is poor, showing a significant gap from current state-of-the-art (SOTA) methods. Specifically, its FPR95 metric on the ImageNet-1k benchmark is only 36.46%, whereas existing methods, such as CLIPN and LSN, have achieved around 30%, and NegLabel has reached 25%. Although the proposed method shows improvements over older baselines, this does not sufficiently demonstrate its effectiveness or make a meaningful contribution to the field. While the authors emphasize that NegLabel uses additional textual information, utilizing a public vocabulary is a standard approach for zero-shot tasks. My personal recommendation is that the authors should validate the method's effectiveness on more recent approaches before resubmitting. In its current form, the paper does not meet the standards for acceptance at ICLR.
2. There appears to be a citation error in Line 519 of the paper; CLIPEN (which might be intended as CLIPN) is not from Dai et al., 2023.

**Questions:**

See Weakness.

---

### Official Review · Reviewer_FJor · 2024-10-30

**Soundness:** 2
**Presentation:** 3
**Contribution:** 2
**Rating:** 3
**Confidence:** 4

**Summary:**

This paper introduces SimLabel, a novel post-hoc strategy that enhances the separability of ID and OOD samples by developing a more robust similarity metric that accounts for consistency across semantically related ID classes. By investigating the image-text comprehension capabilities of VLMs, SimLabel significantly improves OOD detection performance. Extensive experiments across various zero-shot benchmarks highlight its effectiveness, showcasing a promising advancement in achieving robust OOD detection.

**Strengths:**

Strengths：
1. The research contributes to the advancement of zero-shot learning techniques, showcasing how they can be applied effectively to OOD detection scenarios. Besides, it explores the robustness of zero-shot approaches in varied settings, providing insights into how these methods can maintain performance across different data distributions.
2. Despite the performance limitations, the results are presented in a clear and organized manner, allowing readers to understand the implications and significance of the findings within the broader context of the field.
3. The article includes a comprehensive experimental analysis, allowing for a better understanding of the strengths and weaknesses of the proposed method compared to traditional approaches.

**Weaknesses:**

1. The lack of comparison with several state-of-the-art methods that demonstrate superior performance, such as CLIPN [1], LSN [2], NegLabel [3], and CSP [4], represents a significant shortcoming of this study. The relatively low performance is a primary flaw of the article.

[1] CLIPN for zero-shot OOD detection: Teaching CLIP to say no. ICCV, 2023.
[2] Out-of-distribution  detection with negative prompts. ICLR, 2024.
[3] Negative label guided OOD detection with pretrained vision-language models. ICLR, 2024.
[4] Conjugated semantic pool improves OOD detection with pretrained vision-language models. NeurIPS, 2024.

2. It is recommended to include the performance of the proposed method under different CLIP models to achieve a more comprehensive evaluation of its effectiveness.

3. It seems that SimLabel-H and SimLabel-L may not ensure that the similar labels obtained for the ID samples are not from the OOD category. I am unsure if my understanding is correct. If this is indeed the case, I would appreciate it if the authors could provide some clarification regarding the rationale behind the method.

**Questions:**

Please refer to the weakness part.

---

### Official Review · Reviewer_Yowj · 2024-10-30

**Soundness:** 2
**Presentation:** 2
**Contribution:** 2
**Rating:** 3
**Confidence:** 5

**Summary:**

This paper explores the capacity of VLMs for image-text comprehension among closely related ID labels and introduces a novel post-hoc approach called SimLabel. SimLabel strengthens the distinction between ID and OOD samples by establishing a more resilient image-class similarity metric that takes into account the consistency across similar class labels. Extensive experiments validate SimLabel’s superior performance on various zero-shot OOD detection benchmarks, highlighting its effectiveness in achieving robust OOD detection.

**Strengths:**

1. The motivation and paper writing are clear.
2. The experimental analysis is sufficient.
3. The method is simple and easy to implement.

**Weaknesses:**

1. The motivation does not convince me: the ood samples always have the similar appearance with the ID samples. Therefore, their affinity to similar classes should be the approximately the same to ID samples.
2. The OOD example shown in Figure 2 is not the typical sample in current stage, which should be visually similar the ID classes.
3. The  ID dataset in experiment section is limited (only imagenet-1k).
4. The ID accuracy is not provided.

**Questions:**

My main concern is the motivation: See the weakness 1, Why the affinity to similar classes can be helpful to separate the OOD from ID ? I think it can only be helpful to the easy case but not the hard case in OOD detection (visually similar case) ?

---

### Official Review · Reviewer_b5Cd · 2024-10-31

**Soundness:** 3
**Presentation:** 3
**Contribution:** 2
**Rating:** 5
**Confidence:** 5

**Summary:**

This paper investigates the ability of image-text comprehension among different semantic-related ID labels in VLMs and proposes a novel post-hoc strategy called SimLabel.

SimLabel enhances the separability between ID and OOD samples by establishing  a more robust image-class similarity metric that considers consistency over a set of similar class labels.

**Strengths:**

1. The introduced different strategies for generating similar labels is interesting and the authors provide comprehensive insights.

2. The paper learns a robust and discriminative image-class matching score, potentially improving visual classification ability.

**Weaknesses:**

1. Too strong assumption: the work is based on the assumption that  ID sample should consistently have high similarity scores across similar ID classes, which is not often the case, and even worse when encounted datasets with domain gaps.

2. Lack of comparisons and baselines. The comparion of baseline is scarace and the datasets are simple: mcm is not the only post-hoc method and no comparsion with complex OOD setting are displayed.

3. Lack of comparisons and baselines. The comparion of baseline is scarace and the datasets are simple: mcm is not the only post-hoc method and no comparsion with complex OOD setting are displayed.

4. Efficiency: since the method involves generation procedure, e,g. using the external large language models/world knowledge in generating similar classes and so on ( line 156-161),  too many stored knowledge would slow down the efficiecny. I'm curious about the efficiency (real-time throughput and gpu memory consumption) of the method  compared with existing method.

**Questions:**

1. Even MCM serves as the common baseline for zero-shot OOD detection, to my humble understanding, many existing few-shot methods with designed OOD score also suit for post-hoc framework[1, 2, 3], which should also be included for comprehensive coparison.

2. Datasets is not diversity enough. The assumption proposed in line78-79 is not often the case, and even worse when encounted datasets with domain gaps, where OOD datasets are also similar to ID classes. Incorporating more OOD datasets with domian gaps including but not limited to ImageNet-R would strengthen the assumption.

[1] CLIPN for Zero-Shot OOD Detection: Teaching CLIP to Say No. ICCV2023.

[2] Enhancing Outlier Knowledge for Few-Shot Out-of-Distribution Detection with Extensible Local Prompts.

[3] Learning Transferable Negative Prompts for Out-of-Distribution Detection CVPR2024.

---

### Official Review · Reviewer_tfLP · 2024-11-03

**Soundness:** 2
**Presentation:** 3
**Contribution:** 2
**Rating:** 6
**Confidence:** 3

**Summary:**

This paper presents a post-hoc framework for OOD detection with CLIP, SimLabel, which is based on a basic assumption that is an ID sample should consistently have high similarity scores across similar ID classes. The authors introduce three strategies for selecting similar labels. Experiments on several benchmarks show improvement over the baselines.

**Strengths:**

Overall, the motivation and contribution of this work is clearly presented, understandable and consistent with the intuition. Experiment comparison is comprehensive.

**Weaknesses:**

1) In Figure 1 (b), If the model predicts a low probability on other similar categories, then why is the probability high only on the specific category (Egyptian Cat e.g.), and what is the logic behind this?
2) Experimental analysis needs to be added to the experimental results in Table 1, e.g., explanation for the inferior results of this paper's method on the SUN and Textures datasets, and explanation for the difference in results between the three variants.
3) In Figure 4., Why does the FPP increase first when K increases on Places and SUN, while the rest decreases directly?

**Questions:**

Please refer to the above section.

---

### Official Review · Reviewer_uWWf · 2024-11-04

**Soundness:** 3
**Presentation:** 3
**Contribution:** 2
**Rating:** 6
**Confidence:** 4

**Summary:**

This paper introduces a novel post-processing approach called SimLabel, which leverages pre-trained visual-language models (VLMs) for robust anomaly detection. SimLabel utilizes the image-text understanding capabilities of VLMs and emphasizes the consistency of semantic similarity among class labels in the training set, rather than solely relying on predicted similarities for these labels. This approach allows for effective differentiation between in-distribution (ID) and out-of-distribution (OOD) samples. The authors propose three strategies for selecting high-quality, semantically similar class labels: utilizing hierarchical text structures, prompt learning with large language models, and pseudo image-text pairing. Extensive experiments show that SimLabel achieves impressive results across various zero-shot OOD detection benchmarks.

**Strengths:**

- SimLabel harnesses the image-text understanding abilities of pre-trained VLMs, focusing on the semantic consistency among training set class labels rather than relying exclusively on single-class label predictions. This novel approach of leveraging inter-class semantic similarity sets it apart from previous methods, enhancing its capacity to distinguish ID and OOD samples.
- The paper presents three distinct strategies for selecting semantically relevant class labels, including hierarchical text structure utilization, prompt-based learning with large language models, and pseudo image-text pairing. These strategies facilitate the selection of high-quality labels from various perspectives, which strengthens the performance of the SimLabel method.
- The paper clearly explains the underlying principles and implementation details of SimLabel and thoroughly validates its effectiveness through extensive experiments. Results across multiple zero-shot OOD detection benchmarks highlight SimLabel's excellent performance, underscoring the method's advantages.

**Weaknesses:**

- SimLabel’s performance may suffer in long-tail distribution scenarios, where selecting effective similar classes for tail classes becomes challenging. This limitation could reduce its effectiveness in zero-shot OOD detection tasks. One possible improvement could involve expanding tail classes by introducing additional sibling or subclass labels, although careful design would be needed to avoid introducing noise.
- The current implementation of SimLabel assumes equal contribution weights for each similar class when calculating the similarity between images and class labels. In practice, however, the semantic similarity between different classes varies, so a uniform weighting may not capture the full depth of semantic information. Future work could explore refined measures of similarity between classes to improve performance.

**Questions:**

Please refer to the Weaknesses section.

---

### Note · Authors · 2024-11-12

**Comment:**

Thanks for the reviewers and chair.

**Withdrawal Confirmation:**

I have read and agree with the venue's withdrawal policy on behalf of myself and my co-authors.